# ^89^Zr-Immuno-PET with Immune Checkpoint Inhibitors: Measuring Target Engagement in Healthy Organs

**DOI:** 10.3390/cancers15235546

**Published:** 2023-11-23

**Authors:** Iris H. C. Miedema, Jessica E. Wijngaarden, Johanna E. E. Pouw, Gerben J. C. Zwezerijnen, Hylke J. Sebus, Egbert Smit, Adrianus J. de Langen, Idris Bahce, Andrea Thiele, Daniëlle J. Vugts, Ronald Boellaard, Marc C. Huisman, C. Willemien Menke-van der Houven van Oordt

**Affiliations:** 1Department of Medical Oncology, Amsterdam UMC Location Vrije Universiteit Amsterdam, De Boelelaan 1117, 1081 HV Amsterdam, The Netherlands; 2Cancer Center Amsterdam, Imaging and Biomarkers, De Boelelaan 1118, 1081 HV Amsterdam, The Netherlands; 3Department of Radiology and Nuclear Medicine, Amsterdam UMC Location Vrije Universiteit Amsterdam, De Boelelaan 1117, 1081 HV Amsterdam, The Netherlands; 4Department of Pulmonary Diseases, Leiden University Medical Center, Albinusdreef 2, 2333 ZA Leiden, The Netherlands; 5Department of Thoracic Oncology, Netherlands Cancer Institute, Antoni van Leeuwenhoek Hospital, Plesmanlaan 21, 1066 CX Amsterdam, The Netherlands; 6Department of Pulmonary Medicine, Amsterdam UMC Location Vrije Universiteit Amsterdam, De Boelelaan 1117, 1081 HV Amsterdam, The Netherlands; 7Department of Translational Medicine & Clinical Pharmacology, Boehringer Ingelheim Pharma GmbH & Co. KG, Birkendorfer Str. 65, 88397 Biberach, Germany

**Keywords:** ^89^Zr-immuno-PET, target engagement, immune checkpoint inhibitors

## Abstract

**Simple Summary:**

The uptake on a ^89^Zr-immuno-PET scan is not just the result of the binding of a radiolabeled antibody with its target (i.e., target engagement) but also includes background factors such as non-specific binding (for example, catabolism of antibodies inside endothelial cells). In this study, we wanted to isolate target engagement. We used data from five previously performed ^89^Zr-immuno-PET studies with immune-targeting ^89^Zr-radiolabeled antibodies. First, via Patlak analysis, we separated reversible from irreversible uptake, and by using a baseline of target-negative organs, we further defined target-specific irreversible uptake. Second, we compared different mass doses (ratios of labeled and unlabeled antibody) and looked for saturation effects. Evidence for target engagement was based on the following two things: (1) when the target-specific irreversible uptake exceeded the baseline, and (2) when the signal showed saturation. We found target engagement for the different antibodies in several lymphoid organs, for example, in the spleen, while the brain had close to zero target engagement. We propose a new baseline for bone marrow and brain. In conclusion, we promote the use of Patlak analysis for ^89^Zr-immuno-PET studies, or similar simplified outcomes such as a tissue-to-blood ratio.

**Abstract:**

Introduction: ^89^Zr-immuno-PET (positron emission tomography with zirconium-89-labeled monoclonal antibodies ([^89^Zr]Zr-mAbs)) can be used to study the biodistribution of mAbs targeting the immune system. The measured uptake consists of target-specific and non-specific components, and it can be influenced by plasma availability of the tracer. To find evidence for target-specific uptake, i.e., target engagement, we studied five immune-checkpoint-targeting [^89^Zr]Zr-mAbs to (1) compare the uptake with previously reported baseline values for non-specific organ uptake (ns-baseline) and (2) look for saturation effects of increasing mass doses. Method: ^89^Zr-immuno-PET data from five [^89^Zr]Zr-mAbs, i.e., nivolumab and pembrolizumab (anti-PD-1), durvalumab (anti-PD-L1), BI 754,111 (anti-LAG-3), and ipilimumab (anti-CTLA-4), were analysed. For each mAb, 2–3 different mass doses were evaluated. PET scans and blood samples from at least two time points 24 h post injection were available. In 35 patients, brain, kidneys, liver, spleen, lungs, and bone marrow were delineated. Patlak analysis was used to account for differences in plasma activity concentration and to quantify irreversible uptake (K_i_). To identify target engagement, K_i_ values were compared to ns-baseline K_i_ values previously reported, and the effect of increasing mass doses on K_i_ was investigated. Results: All mAbs, except ipilimumab, showed K_i_ values in spleen above the ns-baseline for the lowest administered mass dose, in addition to decreasing K_i_ values with higher mass doses, both indicative of target engagement. For bone marrow, no ns-baseline was established previously, but a similar pattern was observed. For kidneys, most mAbs showed K_i_ values within the ns-baseline for both low and high mass doses. However, with high mass doses, some saturation effects were seen, suggestive of a lower ns-baseline value. K_i_ values were near zero in brain tissue for all mass doses of all mAbs. Conclusion: Using Patlak analysis and the established ns-baseline values, evidence for target engagement in (lymphoid) organs for several immune checkpoint inhibitors could be demonstrated. A decrease in the K_i_ values with increasing mass doses supports the applicability of Patlak analysis for the assessment of target engagement for PET ligands with irreversible uptake behavior.

## 1. Introduction

Monoclonal antibodies (mAbs) targeting immune regulatory checkpoints play a major role in improving anti-cancer treatments in patients, but not all patients benefit, as a significant number only have transient/partial responses or develop resistance [1,2,3]. Personalizing treatment plans to identify patients who will benefit from these treatments is of high clinical value, but the current range of predictive biomarkers is limited and their utility leaves much to be desired. As a well-known example, evaluation of PD-L1 staining on tumour biopsies is currently used to inform clinical decision-making, but factors like intratumoural heterogeneity, interobserver variability, and sampling error can heavily influence the outcome, and PD-L1-negative patients may still respond [4,5,6,7].

With the goal of developing new predictive biomarkers for mAb therapies, we introduced ^89^Zr-immuno-PET: the labelling of monoclonal antibodies with zirconium-89 and their evaluation via whole-body positron emission tomography/computed tomography (PET/CT) [8,9,10,11]. Reasonably so, these ^89^Zr-immuno-PET studies have mainly focused on tumour uptake, but especially in the field of immuno-oncology, whole-body information on healthy (lymphoid) tissue is much needed to understand the spectrum of efficacy and the side effects these mAbs may have through modulating the immune system. Specifically, the target of a mAb-based treatment can be expressed on cells of the lymphoid system and not on the tumour itself. Due to its whole-body information on the mAb biodistribution, ^89^Zr-immuno-PET can be of additional value here.

Besides visualization, PET is a quantitative method to measure the uptake of a radiolabeled mAb. In this regard, it is important to realize that quantification of ^89^Zr-mAb uptake can be inaccurate if only standard uptake values (SUVs) are reported [12]. The SUV contains the sum of all components of uptake in the tissue of interest, including ^89^Zr-mAbs in blood and in the interstitial space, as well as non-specific and target-specific uptake [13]. However, only target-specific accumulation of mAbs is of interest and requires more advanced quantification methods. In addition, the SUV requires stable tracer availability in the blood to be comparable between imaging sessions [14]. Therefore, patient-, treatment-, or mass-dose-specific differences in clearance of the mAb can affect the level of uptake, which needs to be accounted for as well [15,16]. To overcome these issues, Patlak analyses can be performed, enabling the separation of reversible from irreversible ^89^Zr-mAb uptake [17] and accounting for the activity concentration in plasma. In the Patlak equation, the reversible uptake (distribution volume V_T_) of ^89^Zr-mAb comprises the blood volume fraction and unbound antibody distribution, while the irreversible uptake (K_i_) constitutes the target-specific uptake (i.e., the measure of interest) and non-specific uptake (i.e., Fc-mediated uptake and/or pinocytosis [18]).

Two approaches have been introduced previously to further distinguish target-specific from non-specific irreversible uptake. Firstly, Jauw et al. reported baseline K_i_ values (ns-baseline) in non-target expressing organs using biodistribution data from four ^89^Zr- immuno-PET studies, and they demonstrated K_i_ values above ns-baseline representing target-specific irreversible uptake [13]. Secondly, the mass dose is of importance. An adequate imaging mass dose for the evaluation of tumour uptake is defined as a dose high enough to assure tracer availability in the plasma at all time points, but as low as possible to prevent any saturation of tumor targets. Administering higher mass doses, by adding unlabeled mAbs while maintaining the same dose and activity concentration of the radiolabeled mAbs, introduces competition between the radiolabeled and the unlabeled mAbs to bind to the receptor [19]. When the available receptors are limited, this should lead to a decrease in target-specific uptake of radiolabeled mAbs and, subsequently, to a decrease in K_i_. When the mass dose is high enough, receptor binding will be saturated with almost only unlabeled mAbs, while the non-specific irreversible uptake will not be saturated [18].

In this study, employing the ^89^Zr-immuno-PET data for several mAbs targeting immune regulatory checkpoints, we describe evidence of target engagement in healthy (lymphoid) organs by looking at target-specific irreversible uptake and target saturation effects.

## 2. Materials and Methods

### 2.1. ^89^Zr-mAb Studies

Retrospective data from clinical imaging studies performed at our institution meeting the following criteria were used: (1) PET/CT imaging was performed with a ^89^Zr-labeled antibody targeting an immune regulatory checkpoint, (2) a minimum of two PET scans were made at late (>24 h post injection (h p.i.)) time points, and (3) accompanying plasma activity concentration data were available. All clinical imaging studies were approved by the responsible Medical Ethics Review Committee of the Amsterdam University Medical Centers. Patients provided written informed consent. The included studies are shown in Table 1. For further details on study design, we refer to the respective publications.

### 2.2. Biodistribution Analyses

Low-dose CT (ldCT) and PET scans were analysed using the in-house developed BIODISTRIBUTION tool (developed in IDL version 8.4). Based on anatomical location on the ldCT scan, the following organs were manually delineated: brain, kidneys, liver, and spleen. Specifically for liver and spleen, artefacts caused by respiration sometimes occur, causing a mismatch between the ldCT and PET scans. If such a mismatch occurred, the delineation was limited to the area where the ldCT and PET scans overlapped. For the lungs, a standard CT threshold was used for efficiency, followed by manual delineation. The bone marrow was delineated by placing 3 fixed-size regions of interest (ROIs) centrally in 3 separate vertebrae, thereby excluding cortical bone. Mean activity concentrations (Bq/mL) were decay corrected and used as input for further processing.

### 2.3. Blood Sampling

In the respective studies, blood and plasma samples were collected and measured in a cross-calibrated well counter. Data were collected and values are expressed as the activity concentration per liter relative to the injected activity (%IA/L). Time points with more than one observation are presented as the mean and standard deviation, while single observations were visualized with dots. For comparisons of [^89^Zr]Zr-durvalumab, plasma activity concentrations previously published by Verhoeff et al. were extracted and added to the analysis [24].

### 2.4. Patlak Analysis

Activity concentrations in organs and plasma were used for the Patlak analysis [17]. Patlak analysis allows for the separation of reversible and irreversible uptake after equilibrium is reached, which is assumed to be after 24 h for mAbs. The activity concentration in organ tissue (AC_t_) is the sum of these reversible and irreversible parts. The reversible part is proportional to the current supply of mAbs, i.e., the activity concentration in plasma (AC_p_). The irreversible part is proportional to the supply of mAbs accumulated over time, i.e., the area under the plasma curve (AUC_p_). Dividing both sides by AC_p_ gives the linear Patlak equation (see Equation (1)):(1)$$\frac{{AC}_{t}}{AC_{p}}=K_{i}\cdot \frac{A{UC}_{p}}{AC_{p}}+V_{T}$$

The activity concentration data of one region measured at multiple days p.i. can be presented in a Patlak plot, with AUC_p_/AC_p_ along the *x*-axis and AC_t_/AC_p_ along the *y*-axis. Graphical analysis of the Patlak plot provides the slope of the Patlak plot (K_i_), representing the net influx rate of irreversible uptake [h^−1^], and the ordinate intercept, i.e., approaching the total distribution volume (V_T_), as a measure for the irreversible part. K_i_ values were obtained for each organ of each patient. K_i_ values of zero represent an absence of irreversible uptake. Negative K_i_ values were excluded from the analysis, because irreversible uptake cannot decrease over time. As a quality control measure, data with R values of <0.9 for the Patlak linearisation were also excluded from the analysis.

The K_i_ values of four of six organs from the current study were compared with ns-baseline K_i_ values to measure target-specific irreversible uptake. The ns-baseline values are shown in Table 2 [13].

## 3. Results

### 3.1. Finding an Adequate Imaging Dose for ^89^Zr-Immuno-PET Imaging: [^89^Zr]Zr-Durvalumab as an Example

As a first step in ^89^Zr-immuno-PET studies, mass dose selection for optimal visualization and quantification is required. The activity concentrations in plasma relative to the injected activity (%IA/L) from three clinical imaging studies using [^89^Zr]Zr-durvalumab are shown in Figure 1a. For all mass doses, in the first hour after tracer administration, the %IA/L was approximately 30%; most likely, the majority of the ^89^Zr-mAb was still present in the plasma, assuming 2.5–3 L plasma in the body. Interestingly, for the 2 mg mass dose, rapid clearance was noted, with the bulk of the clearance having occurred before the first follow-up time-point (96 h p.i.). Most likely, the clearance occurred sometime in the first 24 h; however, no activity concentration in plasma between 2 and 72 h is available.

The 10 mg mass dose impressively increased the mean activity concentration in plasma by approximately 4 times, i.e., from 2.39 ± 0.70 %IA/L at the 2 mg mass dose to 9.86 ± 8.27 %IA/L at the 10 mg mass dose at 96 h p.i. However, another important issue can be observed, which is the substantial interpatient variability. This is particularly problematic for studies using SUV or %IA/kg as an outcome measure, since the SUV assumes identical clearance across patients and can be heavily influenced by differences in clearance [14].

Finally, the last three doses (22.5, 50, and 750 mg) all seem comparable in terms of clearance and variation, indicating linear pharmacokinetics. To minimize the amount of target saturation, the lowest mass dose, 22.5 mg in this case, would be the most adequate dose. However, a dose between 10 and 22.5 mg could potentially also be appropriate, but no data for this dose range are available. The plasma activity concentration profiles of the other four mAbs investigated are shown in Appendix A. Taking a closer look at the biodistribution of [^89^Zr]Zr-durvalumab, the spleen is one of the organs with major uptake at the 2 mg dose, and the uptake increased to 30 %IA/kg in the first 72 h (Figure 1b). This increase in uptake was not seen with the 22.5 and 750 mg mass doses, where the spleen uptake remained stable and below 10 %IA/kg.

### 3.2. Measuring Target-Specific Irreversible Uptake Using Patlak Analyses

For two patients, one who received 3 mg/kg [^89^Zr]Zr-nivolumab and one who received 200 mg [^89^Zr]Zr-pembrolizumab, one PET scan was missing, resulting in two data points instead of three. For one patient, who received 2 mg [^89^Zr]Zr-durvalumab, PET scan and blood sampling were not performed on the same day; therefore, this time point was excluded, resulting in two data points instead of three. For one patient, who received 750 mg [^89^Zr]Zr-durvalumab, blood sampling and PET were not performed on the same day, and Patlak analysis was not possible. Patlak analyses were performed on the delineated organs to quantify the uptake of [^89^Zr]Zr-mAbs. In all, 132 of 360 K_i_ values were obtained from two data points instead of three data points. To ensure the quality of the data, 31 K_i_ values with R < 0.9 were excluded, and 27 negative K_i_ values were excluded (of which 23 cases received 2 mg [^89^Zr]Zr-durvalumab). After these exclusions, in total, 84% (302 of 360) of the K_i_ values were evaluated (47 of 60 for spleen, 44 of 60 for bone marrow, 57 of 60 for kidneys, 50 of 60 for brain, 50 of 60 for lungs, and 54 of 60 for liver) (see Table 3). Results for the spleen, bone marrow, kidneys, and brain are presented below, and results for the lungs and liver are presented in Appendix A.

#### 3.2.1. Spleen

For [^89^Zr]Zr-BI 754111, [^89^Zr]Zr-durvalumab, [^89^Zr]Zr-nivolumab, and [^89^Zr]Zr-pembrolizumab, a low mass dose resulted in K_i_ values in the spleen above ns-baseline (see Figure 2 and Table 3), indicating target engagement in the spleen. Increasing the mass dose resulted in a decrease in K_i_ values, indicating a saturation effect, with the highest administered mass doses (604 mg for [^89^Zr]Zr-BI 754111, 750 mg for [^89^Zr]Zr-durvalumab, 3 mg/kg for [^89^Zr]Zr-nivolumab, and 200 mg for [^89^Zr]Zr-pembrolizumab) leading to ns-baseline K_i_ values. In contrast, for all mass doses of [^89^Zr]Zr-ipilimumab, ns-baseline K_i_ values were found in the spleen.

#### 3.2.2. Bone Marrow

For every [^89^Zr]Zr-mAb, the lowest mass doses resulted in the highest K_i_ values. The saturation effect was also seen for all [^89^Zr]Zr-mAbs in bone marrow, where the K_i_ values decreased with increasing mass dose (see Figure 3 and Table 3). Though no ns-baseline value was established previously, the highest administered mass dose resulted in similar K_i_ values across the five [^89^Zr]Zr-mAbs. The median K_i_ values ranged from 0.2 to 0.8·10^−3^ h^−1^, suggesting a ns-baseline range for irreversible non-specific uptake in bone marrow.

#### 3.2.3. Kidneys

For [^89^Zr]Zr-BI 754,111 and [^89^Zr]Zr-durvalumab, a low mass dose resulted in K_i_ values in kidneys above the ns-baseline (see Figure 4 and Table 3), indicating the expression of LAG-3 and PD-L1 in kidneys. The three other [^89^Zr]Zr-mAbs showed K_i_ values (partly) overlapping with ns-baseline K_i_ values for all mass doses, which suggests no expression of the target in kidneys. Nonetheless, increasing the mass dose resulted in a decrease in K_i_ values, which is in line with a saturation effect, suggesting an adjusted ns-baseline K_i_ for kidneys from 0.3 to 0.7·10^−3^ h^−1^.

#### 3.2.4. Brain

For all five [^89^Zr]Zr-mAbs, an increase in mass dose did not result in a decrease in K_i_ values, but values remained similar, ranging from 0.0 to 0.1·10^−3^ h^−1^ (see Figure 5 and Table 3). The absence of a saturation effect with increasing mass dose and K_i_ values of almost zero suggest that there is no uptake of these [^89^Zr]Zr-mAbs in the brain and a ns-baseline K_i_ from 0 to 0.1·10^−3^ h^−1^.

## 4. Discussion

Since the first ^89^Zr-immuno-PET clinical imaging trial in 2006 [8], over 30 similar trials have been conducted, spanning 15 different targets, and this number is ever-growing [25]. With the availability of more data, the way to interpret and analyse the results has also evolved. In this study, we showed the ^89^Zr-immuno-PET uptake of five different immune checkpoint inhibitors and demonstrated two approaches of determining target engagement: (1) by measuring target-specific irreversible uptake using a reported ns-baseline uptake value and (2) by increasing the mass dose, resulting in target saturation.

^89^Zr-immuno-PET has the potential to become a promising clinical tool, for example, to better understand the target binding and mechanisms of action of new and existing drugs, but it needs to be interpreted correctly. The potential lies in its ability to non-invasively measure whole-body uptake of a radiolabeled antibody in vivo, with the anticipation of finding clinically relevant correlations. However, the measured PET uptake is not solely a result of target engagement, but also consists of other sources of (background) uptake, such as presence of the ^89^Zr-mAbs in the blood and interstitial space, as well as irreversible uptake in endothelial cells [18]. Distinguishing these components is crucial for the discovery of biologically relevant information and potential correlations to clinical outcomes.

An important first step for every ^89^Zr-immuno-PET study is to establish an adequate mass dose for optimal visualization and quantification. For immune regulatory checkpoints, this is complicated for two reasons: (1) the number of targets in the tumour microenvironment is relatively low [26], and (2) lymphoid organs, such as the spleen, could provide a sink with non-specific uptake that can be highly variable between individuals [15,24]. This is illustrated by the example of [^89^Zr]Zr-durvalumab: on the one hand, a high mass dose can potentially cause saturation of tumour targets, while on the other hand, a lower mass dose causes substantial variation in tracer availability between subjects. The SUV is frequently used to quantify uptake, but it requires comparable plasma activity concentrations across patients and conditions. For this reason, it is essential not only to choose an adequate dose, but also to provide outcome measures that take the activity concentration in plasma into account. Normalization of the activity concentration in plasma can be performed for a fixed point in time (tissue-to-plasma ratios) or by using the area under the plasma curve, which is the approach taken in Patlak analysis [17].

In the spleen, we found target-specific irreversible uptake of [^89^Zr]Zr-BI 754111, [^89^Zr]Zr-durvalumab, [^89^Zr]Zr-nivolumab, and [^89^Zr]Zr-pembrolizumab that was saturated with higher mass doses. This is evidence for target engagement of LAG-3, PD-L1, and PD-1. No target-specific irreversible uptake was found for [^89^Zr]Zr-ipilimumab, suggesting that there is no CTLA-4 expression in the spleen. To explore how these results compare to other tissue-based analyses, we looked for publicly available databases of either protein expression or RNA sequencing. The protein atlas offers a consensus database based on the integration of three RNA sequencing databases: the HPA dataset, the GTEx dataset, and the FANTOM5 consortium [27]. In this consensus database, LAG-3 had a high RNA expression level (23.6), followed by PD-L1 (10.2), PD-1 (7.0), and, finally, CTLA-4 (3.9), suggesting some substantiation of the K_i_ values found (Appendix A), whereas for protein expression, only a single dataset is present based on human annotation, which makes it less reliable than quantitative RNA sequencing data. For protein expression in the spleen, no information is available on CTLA-4 in the protein atlas, and protein expression is recorded as not detected for PD-1, LAG-3, and PD-L1, but there is evidence that the red pulp has high expression of PD-L1 [28]. Therefore, a meaningful comparison of K_i_ values to protein expression was not possible. Apart from the obvious technical differences between RNA sequencing, IHC, and PET imaging, these analyses all represent different derivatives of protein expression. For the K_i_ values reported in this study, it is important to realize that when a residualizing radiolabeled antibody such as ^89^Zr-mAb binds to the target, it is internalized and consecutively degraded intracellularly. During imaging, depending on the target, new receptors can be synthesized and translocated to the cell membrane, probably potentiating the PET signal [15,29]. Therefore, it needs to be considered that K_i_ values rather reflect the in vivo biological availability of the targeted protein over a period of time, rather than a snapshot read-out of protein or RNA expression.

For bone marrow and brain tissue, we propose newly identified ns-baseline values for non-specific irreversible uptake. We estimated the ns-baseline K_i_ for bone marrow to be between 0.2 and 0.8 h^−1^ based on the highest administered mass dose. The K_i_ values in brain were around zero for all [^89^Zr]Zr-mAbs, indicating no irreversible uptake (neither target-specific nor non-specific) in the brain. Presumably, this is because mAbs do not cross the blood–brain barrier of a healthy brain [30]. Of note, the kidneys showed K_i_ values within the ns-baseline for almost all mass dose cohorts. Though the ns-baseline values are considered to represent solely non-specific uptake, there was still a saturation effect present within the ns-baseline range for kidneys. Additionally, the highest mass doses showed K_i_ values even below the ns-baseline. Based on our results, the ns-baseline values for kidneys might actually be more precisely defined and lower than that originally shown by Jauw et al. (2019) [13].

According to the assumptions of the Patlak analysis, the K_i_ value can only be zero or positive, because within the observation time, the irreversible uptake of a residualizing isotope can only remain the same or increase over time. However, we did observe negative K_i_ values, especially in situations with low plasma availability, where the AUC_p_ was most likely overestimated due to the limited sampling time points—an important limitation of this retrospective analysis. To ensure the quality of the data, negative K_i_ values were excluded from the analyses. The majority of the negative K_i_ values were found for the 2 mg mass dose of [^89^Zr]Zr-durvalumab. It is important to note that the included positive K_i_ values for the 2 mg mass dose of [^89^Zr]Zr-durvalumab may be less reliable for the same reason. The few negative K_i_ values found for the other [^89^Zr]Zr-mAbs presumably occurred due to variability in the data.

## 5. Conclusions

In summary, we showed that with ^89^Zr-immuno-PET, we can find evidence for target engagement of immune checkpoint inhibitors, either by using a previously established ns-baseline of non-specific irreversible uptake or by demonstrating target saturation with increasing mass doses, or both. From our findings, we conclude that the PET uptake of radiolabeled mAbs needs to be interpreted in the context of plasma availability of the tracer using appropriate quantitative analyses.

## Figures and Tables

**Figure 1 cancers-15-05546-f001:**
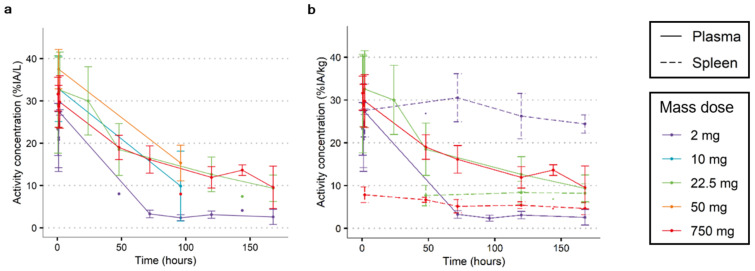
Effect of mass dose on radioactive PK of [^89^Zr]Zr-durvalumab. (**a**) The activity concentration of [^89^Zr]Zr-durvalumab at different mass doses demonstrated that the 2 mg mass dose cleared more rapidly, voiding the plasma after approximately 96 h p.i.. The 10 mg mass dose showed a longer circulation time but high interpatient variability. Finally, the 22.5 mg, 50 mg, and 750 mg mass doses demonstrated linear PK, indicating a preference for the lowest linear mass dose, i.e., 22.5 mg, as an adequate quantification dose for visualization and quantification imaging. For time points with 2 or more observations, the mean %IA/L and SD are shown; additional single observations are shown with single dots. (**b**) With the 2 mg dose, the tracer uptake in the spleen (%IA/kg) demonstrated a rapid initial increase and increased further up to 96 h p.i., coinciding with the decrease in plasma activity. Spleen uptake did not increase over time for the 22.5 and 750 mg mass doses and did not exceed 10% IA/kg at any time point. For time points with 2 or more observations, the mean %IA/L and SD are shown; additional single observations are shown with dots.

**Figure 2 cancers-15-05546-f002:**
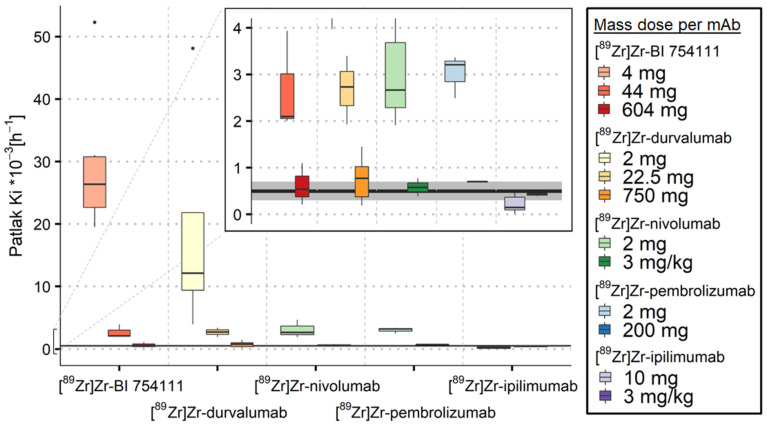
Irreversible uptake in spleen quantified using K_i_ values as a function of mass dose for the five [^89^Zr]Zr-mAbs investigated. Ns-baseline K_i_ median values and interquartile range for spleen are indicated by the black line and grey area, respectively. K_i_ values decreased with increasing mass dose for all [^89^Zr]Zr-mAbs, except for ipilimumab. K_i_ values were comparable to ns-baseline values for high mass doses.

**Figure 3 cancers-15-05546-f003:**
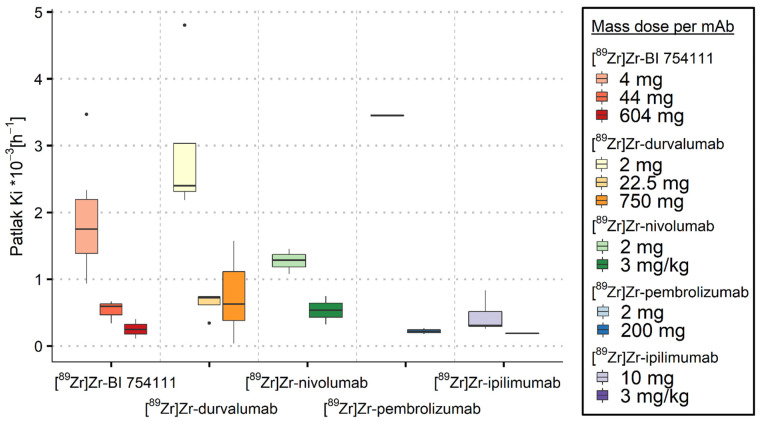
Irreversible uptake in bone marrow quantified using K_i_ as a function of mass dose for the five [^89^Zr]Zr-mAbs investigated. K_i_ values decreased with increasing mass dose for all [^89^Zr]Zr-mAbs.

**Figure 4 cancers-15-05546-f004:**
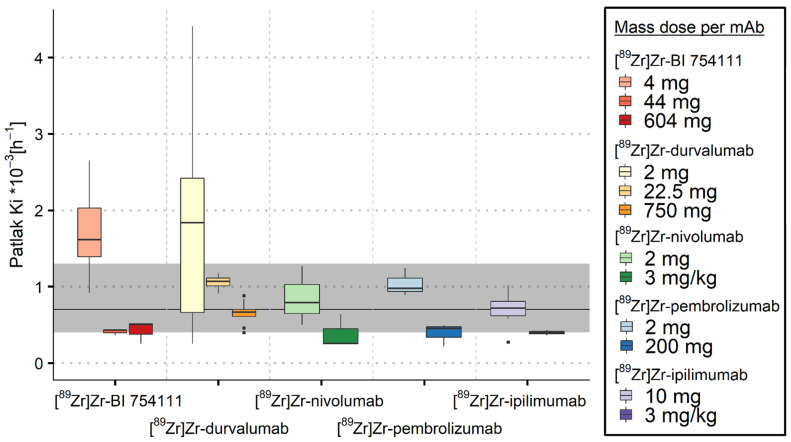
Irreversible uptake in kidneys quantified using K_i_ values as a function of mass dose for the five [^89^Zr]Zr-mAbs investigated. Ns-baseline K_i_ median values and interquartile range for kidneys are indicated by the black line and grey area, respectively. K_i_ values decreased with increasing mass dose for all [^89^Zr]Zr-mAbs. K_i_ values were comparable to ns-baseline values for all mass doses, except for the low mass doses of [^89^Zr]Zr-BI 754111 and [^89^Zr]Zr-durvalumab.

**Figure 5 cancers-15-05546-f005:**
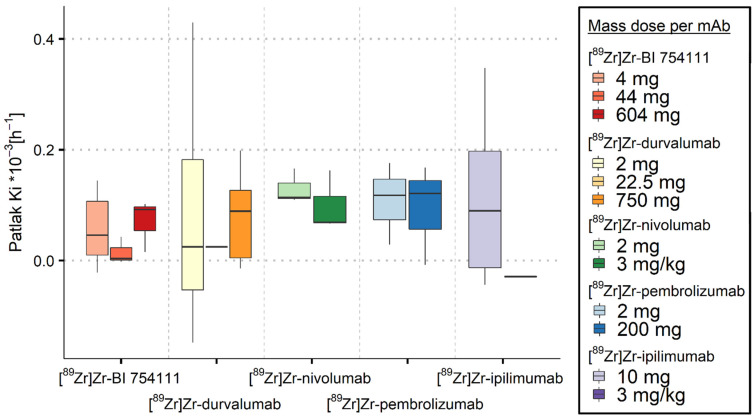
Irreversible uptake in brain quantified using K_i_ as a function of mass dose for the five [^89^Zr]Zr-mAbs investigated. K_i_ values were similar and almost zero for all mass doses of all five [^89^Zr]Zr-mAbs.

**Table 1 cancers-15-05546-t001:** Overview of included ^89^Zr-immuno-PET studies.

^89^Zr-Immuno-PET Tracer	Target	Antibody Isotype	Mass Doses and Scan and Sample Time Points (h p.i.)	N	Tumour Type	References
[^89^Zr]Zr-nivolumab	PD-1	IgG4	2 mg: 1, 72, 120, 168 3 mg/kg: 1, 72, 120, 168	3 3	NSCLC	Niemeijer et al. [20]
[^89^Zr]Zr-pembrolizumab	PD-1	IgG4	2 mg: 1, 72, 120, 168 200 mg: 1, 48, 120, 168	3 3	NSCLC	Niemeijer et al. [21]
[^89^Zr]Zr-durvalumab	PD-L1	IgG1	2 mg: 1, 72, 120, (168) * 22.5 mg: 48, 120, 168 750 mg: 1, 72, 120, (168) *	11 4 10	NSCLC	Smit et al. [22]; Pouw et al., manuscript in preparation
[^89^Zr]Zr-BI 754111	LAG-3	IgG4	4 mg: 2, 96, 144 44 mg: 96, 144 604 mg: 96, 144	6 3 3	NSCLC, HNSCC	Miedema et al. [15]
[^89^Zr]Zr-ipilimumab	CTLA-4	IgG1	10 mg: 72, (96,) 144 ^#^ 3 mg/kg: 72, 96, 144	9 3	melanoma	Miedema et al. [23]

* Eight patients were scanned at two time points 24 h p.i. instead of three according to protocol. ^#^ Four patients were scanned at two time points 24 h p.i. instead of three according to protocol.

**Table 2 cancers-15-05546-t002:** Ns-baseline K_i_ values representing non-specific irreversible uptake reported in [13].

Organ Tissue	Ns-Baseline K_i_·10^−3^ [h^−1^] Median (IQ Range)
Kidney	0.7 (0.4–1.3)
Liver	1.1 (0.8–2.1)
Lung	0.2 (0.1–0.3)
Spleen	0.5 (0.3–0.7)

**Table 3 cancers-15-05546-t003:** Descriptive statistics of irreversible [^89^Zr]Zr-mAb uptake (expressed in K_i_ values) in four healthy organs with five ^89^Zr-immuno-PET tracers.

^89^Zr-Immuno-PET Tracer	Target	Mass Dose	K_i_·10^−3^ [h^−1^] Median (IQ Range)							
			Spleen	N	Bone Marrow	N	Kidneys	N	Brain	N
[^89^Zr]Zr-BI 754111	LAG-3	4 mg	26.3 (22.6–30.8)	6	1.75 (1.38–2.20)	6	1.62 (1.39–2.03)	6	0.05 (0.01–0.11)	6
		44 mg	2.10 (2.05–3.01)	3	0.59 (0.47–0.63)	3	0.43 (0.40–0.44)	3	0.00 (0.00–0.02)	3
		604 mg	0.54 (0.38–0.82)	3	0.25 (0.18–0.32)	3	0.51 (0.38–0.52)	3	0.09 (0.05–0.10)	3
[^89^Zr]Zr-durvalumab	PD-L1	2 mg	12.1 (9.41–21.8) *	4	2.40 (2.31–3.04) *	4	1.84 (0.66–2.42) *	9	0.03 (−0.05–0.18) *	9
		22.5 mg	2.73 (2.33–3.06) *	3	0.72 (0.61–0.74)	4	1.07 (1.01–1.11)	4	0.03 *	1
		750 mg	0.77 (0.38–1.02) *	8	0.63 (0.38–1.12) *	8	0.67 (0.61–0.70)	9	0.09 (0.01–0.13)	9
[^89^Zr]Zr-nivolumab	PD-1	2 mg	2.66 (2.29–3.68)	3	1.29 (1.18–1.37)	3	0.79 (0.65–1.03)	3	0.11 (0.11–0.14)	3
		3 mg/kg	0.58 (0.48–0.67) *	2	0.54 (0.43–0.64) *	2	0.26 (0.26–0.45)	3	0.07 (0.07–0.12)	3
[^89^Zr]Zr-pembrolizumab	PD-1	2 mg	3.21 (2.85–3.29)	3	3.45 *	1	0.98 (0.94–1.11)	3	0.12 (0.07–0.15)	3
		200 mg	0.70 (0.69–0.71)	3	0.22 (0.20–0.24) *	2	0.46 (0.34–0.48)	3	0.12 (0.06–0.15)	3
[^89^Zr]Zr-ipilimumab	CTLA-4	10 mg	0.15 (0.09–0.37) *	7	0.31 (0.30–0.52) *	7	0.72 (0.62–0.81)	9	0.09 (−0.01–0.20 )*	6
		3 mg/kg	0.43 (0.41–0.45) *	2	0.19 *	1	0.40 (0.38–0.41) *	2	−0.03 *	1

* Data excluded from the group due to R < 0.9 or negative K_i_ values.

## Data Availability

Data are available from the corresponding author upon reasonable request.

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
