# Peer review of "89Zr-Immuno-PET with Immune Checkpoint Inhibitors: Measuring Target Engagement in Healthy Organs"

_cancers, 2023, doi:10.3390/cancers15235546_

Round 1

Reviewer 1 Report

Comments and Suggestions for Authors

- Simple summary, is "Elementary Summary" not better ?,

- authors citing, 'saturation', if higher mass of protein are injected, I expect as well 'displacement' of the specific uptake, this is not discussed !,

- in the manuscript; Page 5, patients received up to 3 mg/Kg ?, all the other, mass related injections are indicated in "mg", by, 3 mg/Kg, (pt of 75 Kg), is this 225 mg - while others received 750 mg. This is somewhat confusing !, as 225 mg is less than 750 mg (and 3mg/KG is mentioned in last position)

- in the Simple summary; different mass dose, this is also called 'specific activity', why is this word not used in the whole manuscript ?

Reviewer 2 Report

Comments and Suggestions for Authors

Authors report  analysed data on five  Zr-89  mAbs already used in clinical setting in the Institution in order to delineate target engageemnt  for these different antibodies in several lymphoid organs. U.sing Patlak analyses they separated reversible from irreversible uptake

 Some questions:

 Could the authors specifiy what was the precise timing  for blood and plasma samples?

Inter patient variability is noticed in several studies related to drugs and MRPs pharmacokinetics in plasma and blood due indeed to  patient diffeences in clearance, making SUVs units inadequate for measure and comparisons.  Antibodies are  the spleen and the lymphoid organs as majors organs of uptake.Were there any relations with spleen functional volume or platelets of lymphocyte timing of survival in blood?

How do you explain the variation in  spleen uptake beteween 2 mg mass dose compared to 22.5 and 750 mg?

You describe that in some patients you had  mismatch in PET scan and blood sampling time points , making Patlak analysis not possible. How do explain the mismatch?

In the conclusions you specifiy that  no target specific irreversible uptake was found for Zr -89 ipilimumab suggesting that there is no CTLA-4 expression in the splen.

Was that checkes by immunohistochemistry in any data base ?

Such as The Protein Atlas, for example?

It is interesting to note that Abdies do not cross  the blood brain barrier in healthy brain.

However, elimination viq  kindenys is to be taken in consideration as you talk about saturation effects in the kidenys. However this aspect needs to be analysed and modelized furthermore.

You mentioned several negative Ki values for 2mg mass dose for durvalumab, but also for  22.5 mg and 750 mg  , as well as for nivolumab , pembrolizumab and ipilimumab in Table 3.

Wouldn' t that artefact your interpretation of RP kinetics in spleen and bone marrow.

There is also important variability for the kidney values in Table 3. How do you explain that?

Comments on the Quality of English Language

Good quality of English. The text is easy to read and apprehend

Round 2

Reviewer 2 Report

Comments and Suggestions for Authors

The manuscript was reviewed and authors responded to questions.

Thank you

Drawbacks related to limit in sampls were thoroughly recorded

Comments on the Quality of English Language

OK